# Mitochondrial Protein Quality Control Mechanisms

**DOI:** 10.3390/genes11050563

**Published:** 2020-05-18

**Authors:** Pooja Jadiya, Dhanendra Tomar

**Affiliations:** Center for Translational Medicine, Lewis Katz School of Medicine, Temple University, Philadelphia, PA 19140, USA

**Keywords:** mitochondria, proteome, ubiquitin, proteasome, chaperones, protease, mitophagy, mitochondrial protein quality control, mitochondria-associated degradation, mitochondrial unfolded protein response

## Abstract

Mitochondria serve as a hub for many cellular processes, including bioenergetics, metabolism, cellular signaling, redox balance, calcium homeostasis, and cell death. The mitochondrial proteome includes over a thousand proteins, encoded by both the mitochondrial and nuclear genomes. The majority (~99%) of proteins are nuclear encoded that are synthesized in the cytosol and subsequently imported into the mitochondria. Within the mitochondria, polypeptides fold and assemble into their native functional form. Mitochondria health and integrity depend on correct protein import, folding, and regulated turnover termed as mitochondrial protein quality control (MPQC). Failure to maintain these processes can cause mitochondrial dysfunction that leads to various pathophysiological outcomes and the commencement of diseases. Here, we summarize the current knowledge about the role of different MPQC regulatory systems such as mitochondrial chaperones, proteases, the ubiquitin-proteasome system, mitochondrial unfolded protein response, mitophagy, and mitochondria-derived vesicles in the maintenance of mitochondrial proteome and health. The proper understanding of mitochondrial protein quality control mechanisms will provide relevant insights to treat multiple human diseases.

## 1. Introduction

Mitochondria are double membrane, dynamic, and semiautonomous organelles which have several critical cellular functions. Mitochondria generate energy via oxidative phosphorylation (OXPHOS) and contribute to the metabolism of nucleotides, amino acids, lipids, and cofactors. Mitochondria also regulate calcium homeostasis, ROS signaling, cellular stress responses, and cell death (reviewed in [1]). Thus, mitochondria function as critical players for cellular health and viability. Any dysfunction in mitochondrial homeostasis can lead to the development of numerous human diseases, including neurodegenerative diseases, metabolic syndrome, cardiovascular disorders, myopathies, obesity, type II diabetes, and cancers [2,3,4,5]. A large body of evidence suggests that mitochondria functions depend on the correct mitochondrial proteome. Therefore, it is vital to understand the mechanisms involved in the maintenance of mitochondrial proteome integrity and homeostasis.

Previous evidence supports that mitochondria evolved from the engulfment of an α-proteobacterium by a pre-eukaryotic progenitor cell and comprised of the outer (OMM) and inner mitochondrial membranes (IMM) which invaginate to form the cristae [6]. These phospholipid bilayer membranes separate the intermembrane space (IMS) from the matrix [1,6,7,8]. The mitochondrial membranes have distinct lipid composition, which makes them different from any other membrane present in eukaryotic cells [1,6,7,8]. The phosphatidylglycerol and cardiolipin are exclusively present in the mitochondrial membranes [1,6,7,8]. The protein density in the mitochondrial membranes is higher than other biological membranes; the protein to lipid ratio of OMM is ~50:50, IMM is ~80:20, whereas in the plasma membrane is ~20:80 [9,10,11,12]. Besides that, mitochondrial matrix protein density is around ten-fold higher than the cytosolic protein density [9,10,11,12]. Most of the mitochondrial enzymatic reactions are tightly coupled; therefore, a compact arrangement of the mitochondrial proteome allows its functions to run smoothly. Mitochondria also contain a small circular genome, mitochondrial DNA (mtDNA) that is structured into distinct nucleoids in the matrix and also have their own RNA and protein synthesis machinery [1,7,8]. Studies suggest that mtDNA includes only 37 genes, and most of the mitochondrial genes (~1500) are present at the nuclear DNA (nDNA) [8]. The mtDNA codes for only 13 proteins; the remaining mitochondrial proteins are encoded by nuclear genes [8,13]. The mtDNA encoded proteins are hydrophobic membrane subunits of the electron transport chain (ETC). These proteins synthesized at the membrane-anchored mitochondrial ribosomes, which allow the direct integration of nascent proteins into the IMM [14,15]. The nuclear-encoded mitochondrial proteins are produced on cytosolic ribosomes and actively imported into mitochondria via the TOM/TIM complexes (translocase of outer and inner membrane) [15,16]. The proper coordination between mitochondrial and nuclear genomes is essential for mitochondrial function and perturbations can result in proteotoxicity. Therefore, some mechanisms must exist to protect the mitochondrial proteome from proteotoxicity and to remove damaged proteins or even whole mitochondria, to ensure mitochondrial homeostasis. Experimental evidence suggests that dysfunctional mitochondrial protein quality control is an earlier cellular event involved in several human diseases. However, the understanding of these pathways in human physiology and their association with pathologies remains insufficient.

In this review, we discuss the recent advancements in the field of mitochondrial protein quality control (MPQC) mechanisms in maintaining mitochondrial integrity and function. We mainly focus on the role of chaperones, proteases, ubiquitin-proteasome system (UPS), mitochondria-derived vesicles (MDV), mitochondrial unfolded protein response (UPR^mt^), and mitophagy in assisting correct protein folding, removal of misfolded or aggregated proteins and clearance of dysfunctional mitochondria, thereby ensuring the maintenance of mitochondrial health. Studying the molecular mechanisms of mitochondrial protein quality control will help to understand the pathways for therapeutic targeting in numerous disease conditions.

## 2. Mitochondria Proteome

The landmark paper by Anderson et al., reported the complete sequence of human mtDNA, and recent advancements in large-scale methods including genomics, mass spectrometry (MS)-based proteomics and bioinformatics have defined the mitochondrial proteome in mammals [13,17,18,19,20]. Nevertheless, there are several limitations to identifying all mitochondrial proteins because of their heterogeneity, complexity, tissue specificity, limited specificity of targeting sequence prediction, dual localization, evolutionary origins, and versatility of biochemical methods. However, studies estimate that around 1100–1400 gene loci encode mitochondrial proteins (Table 1) [21]. Hitherto, to date, only about 1500 mammalian mitochondrial proteins have been identified, which include around 300 proteins with unknown functions [13,19,22,23]. The comprehensive compilation of mammalian mitochondrial proteins by Pagliarini et al., known as MitoCarta, suggests 1098 mitochondrial proteins from 14 different mouse tissues [13]. Evidence suggests that approximately half of the mitochondrial proteins are present in all tissues, whereas remaining mitochondrial protein composition varies depending on the tissue type. Besides, the most updated inventory of mammalian mitochondrial proteome MitoCarta2.0 comprises 1158 genes [24]. However, further investigation is required to determine the mitochondrial proteome, which currently remains incomplete. The reader is suggested to check freely available Protein Data Sources including Mitochondrial Protein Atlas (lifeserv.bgu.ac.il/wb/jeichler/MPA), MitoCarta (broadinstitute.org/pubs/MitoCarta) [24], MitoProteome (http://www.mitop2.de) [25], and MitoMiner (mitominer.mrc-mbu.cam.ac.uk) [22], for more detailed mitochondrial proteome information.

Accumulating evidence suggests that mtDNA encodes only a small portion (13 proteins) of the mitochondrial proteome [14,15,26]. The mtDNA encoded proteins are an integral component of the ETC complexes; besides that, two mitochondrial rRNAs and 22 mitochondrial tRNAs are also encoded by the mtDNA [14,15,26]. Previous studies suggest that the OMM is porous and contains about 8–10% of the total organelle’s proteins (~140 proteins), and all are nDNA encoded [27,28,29]. Additionally, around 130 different soluble proteins reported are present in the IMS [30,31]. Based on the available data, the majority of mitochondrial proteins (~800) are localized at IMM, including the multiprotein complexes of ETC [13,23,27]. Furthermore, a previous study using ascorbate peroxidase (APEX)-mediated biotinylation based proteomic mapping identified 495 proteins present in the human mitochondrial matrix [23,27]. All mtDNA and nDNA encoded respiratory chain subunits are involved in the cellular ATP generation via OXPHOS. The IMM comprises multiple large protein complexes ranging from 200 kDa to more than 1 mDa, involved in electron transport named ETC complexes I to IV (C-I, C-II, C-III, C-IV) and ATP synthase; membrane organization complexes named mitochondrial contact site and cristae organizing system (MICOS); protein translocase named translocase of the inner membrane (TIM); multiple proteases and protein processing complexes; ion channels, exchangers and pores like mitochondrial calcium uniporter (mtCU), and mitochondrial permeability transition pore (MPTP) [13,30,32,33,34,35,36,37]. The mitochondrial respiratory complexes I (NADH: Ubiquinone Oxidoreductase) is the largest respiratory complex, comprising of 45 subunits in humans [38,39]. The respiratory complexes are further arranged as supercomplexes (molecular weight ranging from 800 kDa to 5 mDa) in varied stoichiometries of C-I_,_ C-III, and C-IV [40,41,42,43]. The formation of respiratory supercomplexes or respirasomes allows the tight coupling of electron flow and optimizes the use of available substrates [40,41,42,43,44].

Besides the regular mitochondrial proteins, emerging evidence suggests the presence of several microproteins in the mitochondria [45,46,47,48,49,50,51]. The microproteins or micropeptides are peptide molecules encoded by a small open reading frame (sORF) with a length of less than 100 amino acids. Recent reports validated the presence of 20–22 microproteins in the mitochondria [45,51]. The majority of known mitochondrial microproteins are membrane-targeted, present on the OMM [46], or IMM [45,48,49]. The mitochondrial microproteins are known to be involved in the assembly of respiratory chain complexes [45,48], mtDNA translation [47], mitochondrial fatty acid oxidation [49], and endoplasmic reticulum stress response [46]. However, most of the mitochondrial microproteins are functionally orphan and need to be characterized. Therefore, further studies are required to identify correct mitochondrial compartment specificity and assign the molecular functions to the mitochondrial proteins and microproteins.

## 3. Mitochondrial Protein Quality Control System

Mitochondrial proteomic homeostasis is crucial for cellular health and bioenergetics. The correct protein import, targeting, folding to functional proteins, and their regulated turnover are essential for mitochondria health and integrity. All these molecular processes are collectively termed as mitochondrial protein quality control (MPQC) system (Figure 1). The failure or dysregulation of the MPQC results in mitochondrial dysfunction that can cause and contribute to the development of various diseases. The nDNA encoded proteins are synthesized either at the OMM associated ribosomes or the ER-associated ribosomes [52,53]. There are three major sites for the MPQC: (1) at the OMM, which is associated with the mitochondrial protein import machinery or the whole organelle turnover through mitophagy; (2) in the IMS, which is involved in the efficient targeting of the membrane proteins and turnover of OMM, IMS, and IMM targeted proteins; and (3) in the matrix, which is responsible for the turnover of misfolded, unfolded, or matrix and IMM localized proteins.

### 3.1. Mitochondrial Protein Import and Mitochondria-Associated Degradation Systems

The majority of nDNA encoded mitochondrial proteins (~60%) have specific N-terminal mitochondrial-targeted sequences, termed as presequence. The presequence containing mitochondrial preproteins are imported into the mitochondria through the well-characterized classical protein import system present at the OMM and IMM, named the translocase of the outer membrane (TOM), and translocase of the inner membrane (TIM) [52,53,54]. The TOM complex forms a dimeric or trimeric structure, which is constituted by multiple subunits, channel-forming β-barrel protein TOM40, and α-helical membrane-integrated subunits (regulatory components TOM5, TOM6, and TOM7; receptor subunits TOM20, TOM22, and TOM70) [54,55,56]. Mitochondrial preproteins are recognized by the TOM20 and TOM22 and translocated across the OMM through TOM40. At the IMM, the preprotein is recognized by the receptor component TIM50, and mitochondrial membrane potential (Δψ) facilitates translocation of positively charged presequence through the pore-forming components TIM23 and TIM17 [54,55]. The presequence translocase-associated motor (PAM), ATP-dependent mitochondrial heat-shock protein 70 (mtHSP70), mitochondrial processing peptidase (MPP), Mitochondrial genome-required protein 2 (Mgr2)/Reactive oxygen species modulator 1 (ROMO1), and inner-membrane peptidase (IMP) are involved in the release of the preprotein into the matrix, IMM, or IMS [53,54,55].

Besides the classical import system, mitochondrial proteins could be transported through other import systems, which recognize different kinds of internal targeting signals [53,54]. The β-barrel proteins of the OMM like Voltage-dependent anion channels (VDACs) and TOM40 have a characteristic targeting signal that consists of a β-hairpin element recognized by the TOM complex. The β-barrel proteins are inserted into the OMM with the help of the TIM9/TIM10 chaperone complex, and the sorting and assembly machinery (SAM; composed of SAM50, SAM37, and SAM35) present at the OMM [53,57]. The α- helical transmembrane proteins of the OMM can be categorized into three main classes: signal-anchored proteins, tail-anchored proteins, and polytopic (multispanning) outer-membrane proteins. The α- helical transmembrane proteins can be inserted into the OMM through multiple routes such as the mitochondrial import (MIM) complex, or via a coordinated action of the TOM/TIM and the MIM complex together, or in a lipid-assisted manner [53,54]. Some IMS proteins have a characteristic cysteine-rich region, and these proteins are imported by TOM and recognized by the mitochondrial IMS import and assembly (MIA) system. The MIA40/CHCHD4, which is a central component of the MIA system, recognizes the cysteine-rich region and forms a transient intermolecular disulfide bond [58]. The sulfhydryl: cytochrome C oxidoreductase GFER/ERV1 subsequently reoxidizes the MIA40 through a disulfide relay system [59]. The hydrophobic metabolite carrier proteins of the IMM are usually bound to cytosolic chaperones, like HSP70, and lack a precursor sequence [53,54]. The TOM70 recognizes and binds to both chaperone and metabolite carrier proteins and subsequently translocated carrier protein to the IMS through the TOM40 [53,54]. In the IMS, a small chaperone complex TIM9/10 recognizes the carrier protein and forms a carrier protein-TIM9/10/12 complex which is recognized by the TIM50. Further, TIM22 facilitates the insertion of the carrier protein to IMM with the help of TIM18 and SDH3 [53,54]. The mitochondrial import system is well characterized in the yeast, and most of the mammalian homologs of the TOM/TIM complex are also identified.

The TOM complex at the OMM is a major entry site for the mitochondrial proteins and is involved in the majority of protein import pathways. The translocation of mitochondrial-targeted nascent proteins is tightly coupled with the mitochondria-associated degradation (MAD) system present at the outer face of the OMM through the TOM complex. Recently, multiple pathways for the MAD have been discovered, namely, VMS1/VCP/MDM30 mediated [60], mitoRQC [60,61], MAGIC [62], mPOS [63], UPRam [64], mitoCPR [65], and mitoTAD [66] (Table 2). The turnover and quality control of the OMM proteins can be ensured by the cytosolic ubiquitin-proteasome degradation system (UPS). The OMM protein Mitofusin (MFN/Fzo1) is a well-known target for the cytosolic UPS [67]. The MFN is ubiquitinated by F-box protein Mdm30, which associates with Skp1-Cullin-F-box (SCF) ubiquitin ligases complex [67]. The Velocin-containing protein (VCP) associated Mitochondrial Stress-responsive 1 (VMS1) was discovered as a cytosolic sensor of mitochondrial stress [60]. In response to mitochondria stress, MDM30 ubiquitinates OMM proteins, namely MFNs [60]. Subsequently, VMS1 translocates to the mitochondria and recruits the ER-associated degradation system (ERAD) component Cdc48/VCP/p97. This Cdc48/VCP/p97 consequently recruits the UPS machinery for the effective degradation of the mitochondrial proteins and protects the mitochondria against the stress condition [60]. Recently, VMS1 has been shown to establish a link between the cytosolic ribosome quality control (RQC) and the mitochondria, and this pathway is termed as mitoRQC [61]. The RQC is involved in the turnover of misterminated nascent proteins, which are stuck at the ribosomes [68,69]. The RQC machinery detects the stalling of translation, which could be due to defective mRNA or aberrant translation [68,69]. The ribosomal protein Rqc2/NEMF and a ubiquitin E3 ligase Ltn1/Listerin are the central components of the RQC. The Rqc2/Ltn1 are involved in the detection and ubiquitination of the misterminated nascent proteins which are subsequently degraded by the proteasomes [68,69]. The ribosomes are in proximity to the TOM complex, which allows the translation coupled translocation of the nascent mitochondrial proteins. At the ribosome, Rqc2 recruits Ltn1 for the ubiquitination and subsequent degradation of the stalled peptides. Vms1 binds directly to OMM associated ribosome to antagonize the Rqc2 mediated Ltn1 recruitment, which allows the translocation of the stalled peptide to IMS [61]. In the IMS, the mistranslated peptide can be degraded by the mitochondrial proteases [61]. Therefore, Vms1 and Ltn1 are the critical components of the mitoRQC. In the absence of both Vms1 and Ltn1, the stalled peptide can aggregate in the mitochondria due to the Rqc2 mediated addition of the C-terminal alanyl/threonyl sequences (CAT-tails) [61]. Besides the nascent mitochondrial proteins, the cytosolic proteins which are prone to aggregation are shown to be imported into the mitochondria and subsequently degraded through the mitochondrial proteases [62]. This mitochondria-associated proteostasis (protein homeostasis) system is termed as “mitochondria as guardian in cytosol” (MAGIC) [62]. The cytosolic HSP70s facilitate the mitochondrial import of the misfolded proteins through the TOM/TIM complex, and inside the mitochondria, misfolded proteins are degraded by the Lon protease [62].

The aberrant mitochondrial precursor protein accumulation in the cytosol activates a stress response termed as “mitochondrial precursor over-accumulation stress” (mPOS) [63]. Mistargeting of proteins to the mitochondria also induces a stress response named “unfolded protein response activated by mistargeting of proteins” (UPRam) [64]. Both mPOS and UPRam initiate a signaling cascade, which subsequently reduces the global translation and activates UPS mediated protein degradation [63,64]. The inhibition of the mitochondrial protein import could also activate a stress response termed as “mitochondrial Compromised Protein Import Response” (mitoCPR) [65]. The mitoCPR increases the expression of Cis1 (OMM protein) through the activation of the transcription factor PDR3 [65]. The Cis1 interacts with TOM70 and recruits the AAA-ATPase Msp1/ATAD1 for the release of mitochondrial preproteins from the TOM complex and makes them available for UPS mediated degradation [65]. The Msp1/ATAD1 is involved in the degradation of the mislocalized tail-anchored proteins at the OMM [70,71]. Mitochondrial stress, which disrupts the mitochondrial import pathways, could lead to the clogging of the TOM complex. Recently, a protein quality control system that involves the ERAD component Ubx2, named “mitochondrial protein translocation-associated degradation” (mitoTAD), has been discovered [66]. The mitoTAD allows the removal of the halted proteins and prevents the clogging of the TOM complex. The Ubx2 was identified as an interacting partner of the pore-forming subunit TOM40 using an affinity purification approach [66]. Ubx2 is known to be involved in the ERAD, where it acts as an adaptor for Cdc48/VCP/p97 and facilitates UPS mediated protein degradation. Similarly, in the mitoTAD pathway, Ubx2 senses the stalled preproteins in the TOM complex and recruits the Cdc48/VCP/p97 to extract the jammed preproteins and destine them to the UPS mediated degradation [66]. Therefore, mitochondria have a sophisticated protein quality control system that intimately works with the mitochondria protein import system. These MPQCs allow the detection and removal of the mistargeted, stalled preproteins at the translocases, along with the activation of the global proteostasis response. Most of these MPQCs were discovered in lower organisms (yeast model system) and have not been studied in mammalian systems, thus they require further investigation.

Ubiquitination plays an essential role in the OMM protein turnover and MPQCs associated with the protein import system. However, the ubiquitination of intramitochondrial proteins is not well characterized and still debatable. A few recent reports suggest that intramitochondrial protein could be ubiquitinated. Using mass spectrometry, Ciechanover and colleagues showed the presence of ubiquitinated proteins in isolated mitochondria and suggested that ~62% of the mitochondrial proteome might be ubiquitinated [72]. They further validated the presence of a ubiquitin E3 ligase Dma1p/RNF8 in the mitochondria, which was previously identified as a nuclear localizing E3 ligase involved in the nonhomologous end-joining (NHEJ) DNA damage repair [72,73]. The process of ubiquitination (conjugation of ubiquitin-protein to the lysine residues in a target protein) involves the sequential action of three enzymes; ubiquitin-activating enzyme E1, conjugating E2, and ligase E3 [74]. The E3 ligases provide the selectivity to the ubiquitination by identifying and recruiting the substrate protein [74]. Experimental evidence suggests the presence of the ubiquitination enzymatic machinery in the mitochondria. The ubiquitin-activating enzyme E1 was identified in the mitochondria using quantitative electron microscopic immunolocalization [75]. Recently, the E2 enzyme Ubc9 was also identified in the mitochondria [76]. Multiple proteomic-based experimental data suggest that the mitochondrial proteome is ubiquitinated [72,77,78,79]. Recently, Lavie et al. demonstrated that 203 mitochondrial proteins are ubiquitinated and ~82 of these ubiquitinated proteins are localized either in the IMM or the matrix [77]. The turnover of several OXPHOS proteins is regulated by UPS [77]. Using a chimeric HA-tagged ubiquitin coupled to a mitochondrial leader sequence (mt-Ubi), the authors also validated that ubiquitination occurs inside the mitochondria [77]. These experimental data suggest that intramitochondrial ubiquitination could be an essential MPQCs mechanism, which needs to be further explored.

### 3.2. Chaperone-Mediated Quality Control

The nDNA encoded polypeptides or mitochondrial preproteins are imported to an intramitochondrial destination (see “Protein import” above). During the import process, precursor proteins need to be in an unfolded state due to the small dimensions of the translocation channels. After reaching the target compartment, imported proteins refold to become functional and achieve their native conformation. This mitochondrial protein translocation is efficiently carried out with the help of mitochondrial chaperones, commonly known as heat shock proteins (Hsp), that serve many important roles throughout the process. The mitochondrial chaperones assist in preprotein translocation across the mitochondrial membranes, unfolding of preproteins, identification of unfolded polypeptides, folding of newly imported proteins inside the matrix, degradation of misfolded proteins to prevent prion formation and disaggregation of protein aggregates [27,53]. Increasing evidence indicates that mitochondria encompass a complete set of molecular chaperones or Hsp, independent of the cytosolic Hsp that are located in the IMS and matrix [16]. The leading chaperone families include Hsp40, Hsp60, Hsp70, Hsp90, Hsp100/Clp, and sHsp (small Hsp) that are named by their estimated molecular mass [80]. The role of Hsp60 and Hsp70 families has been extensively studied and it has been shown that these families play an important part in MPQC [81].

The Hsp70 chaperone known as mtHsp70, Grp75, or mortalin in humans, is ubiquitously expressed, localized in the matrix, and encoded by the HSPA9 gene. The mtHsp70 performs multiple functions such as polypeptide chain translocation through the TOM/TIM translocase complexes, unfolding of the preprotein and folding of imported proteins, biogenesis of iron-sulfur clusters in the matrix, and protein stabilization [82,83,84]. Hsp70 contains three domains: (1) an N-terminal nucleotide-binding (ATPase) domain that binds ATP and hydrolyzes it to adenosine diphosphate (ADP); (2) a substrate/polypeptide binding domain; and (3) a C-terminal domain that is rich in an α-helical structure and works as a ‘lid’ for covering the substrate-binding domain [85]. The substrate (peptide)-binding affinity of Hsp70 chaperones depends on their intrinsic ATPase activity and demonstrates low substrate-binding affinity in the ATP-bound state and high substrate-binding affinity in ADP-bound state after ATP hydrolysis [86]. Two types of conserved co-chaperones control the ATPase activity of Hsp70 chaperones; the DnaJ-like or J-domain proteins, and a nucleotide exchange factor (NEF) [86]. Previous evidence demonstrated that J-domain proteins help in initial polypeptide interaction, binding, and stimulation of the ATPase activity of Hsp70. However, J-domain proteins are also considered as a distinct family of a chaperone, termed Hsp40 [87,88]. In humans, there are different types of J-domain containing proteins, DNAJA3/Tid1, DNAJC19, and DNAJC20 (HSCB, HSC20) [89]. On the other hand, NEF, a member of the GrpE protein family (GRPEL1, GRPEL2), initiates substrate release by stimulating the exchange of ADP for ATP [86].

The Hsp60 family members, also known as chaperonins, which were the first identified molecular chaperones in the matrix, have an essential role in the biogenesis process of cellular proteins [90,91]. The Hsp60s family comprises Hsp60 and its cofactor Hsp10 subunits making a barrel-shaped complex that is primarily involved in the folding of relatively small, imported polypeptides and assists in the refolding of misfolded substrate proteins into the properly folded state [81]. A proteomic study reveals that ~30% of mitochondrial proteins are folded with the help of Hsp60/Hsp10 (chaperonin) complex [92]. In addition, the mitochondrial Hsp90 known as TRAP1 (TNF receptor-associated protein 1) is matrix localized and believed to assist in protein folding [93,94]. TRAP1 has been shown to protect against oxidative stress, apoptosis, and MPTP opening [95]. Members of the Hsp100/Clp protein chaperone family mostly mediate unfolding and disaggregation of insoluble misfolded proteins. Hsp78 or ClpB, a member of the Hsp100/Clp protein chaperone family, is involved in the disaggregation of the proteins that cannot be removed by proteolytic systems [96]. In mammals, a putative mitochondrial ClpB homolog (CLPB) has been identified [97]. However, its specific function in mitochondrial proteome homeostasis has not been studied.

### 3.3. Proteases Involved in MPQC

Mitochondrial proteases or mitoproteases are an integral component of the MPQC and responsible for the protein turnover and processing inside the mitochondria. There are around 45 mitoproteases known in mammals, which can be categorized into two major classes; resident and transient [98,99]. The resident mitoproteases (~25) solely reside in the mitochondria and are involved in the regulation of nearly all mitochondrial functions. The resident mitoproteases can be localized to the matrix, IMM, IMS, or OMM. These resident mitoproteases can be categorized into three major catalytic classes: metalloproteases, cysteine proteases, and serine proteases (Table 3). The resident mitoproteases play a key regulatory role in protein trafficking, processing, activation, and maintenance of the mitochondrial proteome by regulating the protein turnover [98,99]. As nDNA encoded mitochondrial proteins are imported from the cytosol through the mitochondrial protein system, three protease systems (MPP, IMP, and MIP) ensure the proper trafficking and processing of nascent proteins. Mitochondrial targeted presequence containing proteins are processed in the matrix by the MPP protease complex (PMPCA and PMPCB) [16,53,55,99]. A few preproteins undergo additional processing after the removal of their presequence, which is carried out by MIP, or by XPNPEP3 [16,53,55,99]. Several proteins are translocated to the IMM or IMS after processing with MPP. However, most of mtDNA encoded proteins, are processed by IMP (IMMP1l, IMMP2L) [16,53,55,99]. mAAA proteases also process some nascent matrix proteins. The METAP1D protease is involved in the N-terminal methionine excision of some mitochondrially encoded polypeptides [100].

Besides protein import and trafficking, multiple proteases are involved in the homeostasis of mitochondria proteome. These proteases work intimately with the chaperones to fine-tune the mitochondrial proteome according to cellular demands. The LON and CLPP (AAA proteases) are the two major players for the removal of damaged matrix proteins [98,99]. At the IMM, two AAA protease complexes (mAAA and iAAA) ensure the majority of protein quality control. The catalytic domain of the mAAA protease system (AFG3L2, SPG7) is facing the matrix and involved in regulating the ETC complexes, mitochondrial calcium uniporter complex, and mitochondrial translation [99,101,102]. The catalytic domains of the i-AAA protease system (YME1L, OMA1, ATP23) are facing the IMS. ATP23 is involved in mitochondrial phospholipid metabolism and maintenance of ETC complexes [98,99,103]. The OMA1 and YME1L modulate mitochondrial dynamics through the OPA1 proteolysis [99,104,105]. HTRA2 is involved in the degradation of the damaged IMS proteins besides its role in apoptosis [99,106,107]. The peptides generated through the proteolysis can be further degraded into amino acids by the protease PITRM1 [99]. The USP30 present at the OMM is a deubiquitinating enzyme and protects the mitochondria from aberrant mitophagy by antagonizing the PINK1/PARKIN induced ubiquitination [108]. Besides the resident mitoproteases, mitochondria also house several transient mitoproteases (~20), which are involved in multiple mitochondrial processes [98]. However, the majority of transient mitoproteases are not well explored and further study is needed to establish their role in mitochondrial physiology and functions.

### 3.4. Mitophagy and Mitochondria-Derived Vesicles in MPQC

At times, the above-discussed MPQC mechanisms that operate mainly on the protein levels, become defective or exhausted. Failure of these mechanisms can cause accumulation of denatured polypeptides and protein aggregates that would create a severe burden to mitochondrial functions. As mentioned, defects in mitochondrial functions can cause the generation of toxic chemicals such as ROS and can cause cell death via different mechanisms, including apoptosis and opening of MPTP. Thus, cells have an alternative mechanism to eliminate damaged mitochondria selectively and to preserve the mitochondrial functional quality, termed as mitophagy. In general, autophagy includes the formation of double-membrane vesicles known as an autophagosomes in which the sequestration of the cytoplasmic material, including organelles, cytosolic proteins, or protein aggregates, occurs. The autophagosome fuses with a lysosome and forms an autophagolysosome in order to trigger degradation of the sequestered cytoplasmic material [109,110]. Autophagy also acts as a robust recycling system of the intracellular components and provides energy for cellular renovation and homeostasis [111,112,113]. A detailed mechanism of autophagy has been reviewed elsewhere [109,110]. While non-selective autophagy occurs in response to nutrient deprivation, mitophagy is the mitochondria-specific highly selective autophagy that removes mitochondria either to degrade defective ones or to control their number to match metabolic demand by a specific mechanism.

Sam L. Clark, Jr. uncovered for the first time the presence of mitochondria within an autophagosome in 1957 [114]. Since then, autophagy has been thought of as one of the important pathways for the degradation of mitochondria. Recently, several studies supported the role of mitophagy in the maintenance of mitochondrial quality control in both yeasts and mammals [115,116,117]. Two different groups independently identified mitophagy related genes through genetic screens of mitophagy-defective mutants in yeast *Saccharomyces cerevisiae*, which suggests that mitophagy shares some core autophagic machinery with non-selective autophagy [118,119]. However, selectivity for mitochondrial cargo degradation is mediated by a unique set of proteins.

Mitophagy in mammals is mediated by two different pathways: receptor and ubiquitination mediated. Receptor-mediated mitophagy in mammals shares some resemblance with yeast mitophagy. In yeast, it is mediated by a mitophagy-specific receptor Atg32 (autophagy-related 32), present at the OMM, which binds to the Atg8 and Atg11 [118]. This binding allows the recruitment of mitochondria to the autophagosome [118]. A mammalian homolog of Atg32 has not been identified yet. However, some mitophagy receptors have been reported, which are functional counterparts of the Atg32. These receptors include NIX (NIP3-like protein X), also known as BNIP3L (BCL2/adenovirus E1B-interacting protein 3-like), Bcl2L13 (Bcl2-like 13), FUNDC1 (FUN14 domain-containing protein 1), and BNIP3 (BCL2/adenovirus E1B 19-kDa-interacting protein 3) [120,121,122,123]. NIX-mediated mitophagy is involved during red blood cell differentiation to remove intact mitochondria [122]. BNIP3 is involved in hypoxia-induced autophagy and also activates mitophagy [123,124]. All these receptors are outer-mitochondrial membrane proteins and have an LC3-interaction region (LIR) that mediates binding of these receptors to LC3 (the mammalian Atg8 orthologue). Together, receptor-mediated mitophagy, also known as the PARKIN-independent pathway, results in the recruitment of lipidated LC3 proteins to damaged mitochondria in an LIR-dependent manner [120,121,122,124]. However, the role of these receptors in the regulation of mitophagy in different physiological conditions is less characterized and thus needs further validation.

The ubiquitination-mediated pathway is known as PINK1 (phosphatase and tensin homolog (PTEN)-induced kinase)- and PARKIN-dependent mitophagy. Mutations in both genes are linked with familial Parkinson’s disease [125,126]. Early studies in *Drosophila* models linked PINK1 and PARKIN to many mitochondrial functions [127,128] and supported the idea that PINK1 and PARKIN act together and PINK1 function upstream of PARKIN [129]. PINK1 and PARKIN both regulate mitochondrial quality controls and functions such as mitochondrial transport, movement, dynamics, and mitophagy [130,131,132]. Studies show that PARKIN (an E3 ubiquitin ligase) resides in the cytosol, but during mitochondria dysfunction, it is recruited from the cytosol to depolarized mitochondria [132]. PINK1 has a highly conserved C-terminal serine/threonine kinase domain homologous to the Ca^2+^/calmodulin family and a predicted mitochondrial targeting sequence (MTS) at the N-terminus, suggesting that it is localized in mitochondria [126]. Under healthy conditions, PINK1 is imported into mitochondria via TOM/TIM complexes, and MTS is cleaved off by the MPP and PARL [133,134]. Cleaved PINK1 is subsequently released into the cytosol and targeted for degradation by the proteasome [135]. The defective mitochondria are unable to import and degrade PINK1, so it accumulates on the outer mitochondrial membrane (OMM), where it phosphorylates both PARKIN and ubiquitin [134,136,137]. PARKIN phosphorylation is a signal to activate its E3 ligase activity, which leads to the ubiquitination of the VDAC, TOM20, MIROs, MFNs, and multiple other OMM proteins to label the damaged mitochondria for mitophagy [132]. Autophagy adaptors mediate the interaction between ubiquitinated mitochondrial proteins and LC3, which allows selective engulfment of ubiquitinated mitochondria by the autophagosome. In mammals, five major adaptor proteins have been linked to mitophagy: optineurin (OPTN), nuclear domain 10 protein 52 (NDP52), neighbor of BRCA1 gene 1 (NBR1), TAX1 binding protein 1 (TAX1BP1), and p62 [138]. However, it is not fully clear how PINK1 recognizes damaged mitochondria. Emerging evidence suggests that the loss of Δψ is a signal for PINK1. Under basal conditions, full-length PINK1 protein import into mitochondria occurs in a Δψ dependent manner [133]. On the contrary, PINK1 can also accumulate on the OMM, independent of the Δψ [139], suggesting some other mechanism of mitophagy activation might exist. Taken together, this pathway is well characterized; however, further investigation is still needed for a complete understanding of the mitophagy signaling.

Recently, mitochondria-derived vesicles (MDVs) were discovered as a means of MPQC, where cells cannot afford to lose large mitochondrial content due to mitophagy. The size of MDVs ranges from 70–150 nm, and these can be generated from OMM only, or both the OMM and IMM [140,141]. Depending on membrane content, MDVs might carry the OMM and IMS content or OMM, IMS, IMM, and matrix contents. However, the content of MDVs also depends on the stress condition [140,141]. The generation of MDV does not require mitochondrial fission regulator Drp1 [142]. The MDV can deliver the mitochondrial content to the multi-vesicular body (MVB) pathway, peroxisomes, or lysosomes [140,141,142,143]. The lysosomal delivery of the MDVs leads to the degradation of the mitochondrial content [140,143]. The regulatory events associated with the MDVs generation and involvement of the MDVs in pathophysiology is not well established and therefore needs further studies.

### 3.5. Mitochondrial Unfolded Protein Response (UPR^mt^)

Mitochondrial quality control and maintenance of mitochondrial proteostasis are supported by the UPR^mt^, a mitochondria-to-nuclear stress signal transduction pathway. Evidence suggests that mitochondria have protein-folding mechanisms that include different chaperones and proteases, as discussed above, to lessen mitochondrial proteotoxicity. These mitochondrial chaperones (mtHSP70 and HSP60–HSP10) are present in the matrix and required for protein import and to promote proper protein folding. Similarly, proteases present in the matrix and IMM are required for the degradation of misfolded proteins. UPR^mt^ is considered as a stress response that activates gene transcription of nuclear-encoded mitochondrial chaperones and proteases in order to enhance protein homeostasis. Although this pathway was first discovered in mammals [144], the molecular mechanism has been comprehensively studied in *Caenorhabditis elegans* [145]. This section discusses the UPR^mt^ mechanisms in both *C. elegans* and mammals.

Yoneda et al. generated a *C. elegans* model to activate the UPR^mt^ pathway by expressing GFP under the control of mitochondrial chaperones hsp-6 and hsp-60 gene promoters (homologs of mammalian mitochondrial matrix chaperones mtHSP70 and HSP60, respectively) [145]. In this study, the authors demonstrated that basal levels of hsp-6 and hsp-60 derived GFP expression increase after treatment with ethidium bromide [145]. Here, authors used ethidium bromide as a first UPR^mt^ stressor in worms to induce mitochondrial stress, which is known to reduce mtDNA transcription and replication [146]. Subsequently, many RNAi-based screens were performed using the UPR^mt^ reporter *C. elegans*, which led to the discovery of multiple UPR^mt^ components. These include: ubl-5, a ubiquitin-like protein; DVE-1 (defective proventriculus), a homeodomain-containing transcription factor; clpp-1, an ATP-dependent mitochondrial matrix protease (homologous to CLPP); and haf-1, an inner membrane-spanning ATP-binding cassette (ABC) peptide transporter protein [147,148]. It is believed that UPR^mt^ signaling starts when unfolded, misfolded or unassembled proteins accumulate and aggregate in the mitochondrial matrix beyond the folding capacity of mitochondrial chaperones. The ClpP protease degrades these protein aggregates into short peptides (6–30 aa. in length) [147]. Then, these peptides are transported by HAF-1 from the mitochondrial matrix to IMS and then cross the OMM to diffuse into the cytosol [148]. However, these peptides themselves can interrupt mitochondrial protein import. Consequently, HAF-1-mediated peptide efflux activates and causes the nuclear translocation of the bZip transcription factor ATFS-1 (activating transcription factor associated with stress-1, previously known as ZC376.7), the primary UPR^mt^ regulator [149]. In the nucleus, ATFS-1, in conjunction with the second transcriptional complex, including Ubl5 and DVE-1, starts a transcriptional program to activate the expression of mitochondrial chaperones, proteases, glycolytic proteins, genes linked with mitochondrial biogenesis, ROS detoxification and mitochondrial protein import to restore protein homeostasis [149]. Haynes and colleagues discovered that this transcription factor has a mitochondrial targeting sequence (MTS) along with its nuclear localization sequence (NLS) [149]. The study revealed that in healthy mitochondria, ATFS-1 is imported into the mitochondrial matrix and degraded by the LON protease [149]. However, because of mitochondrial dysfunction and impaired mitochondrial protein import, ATFS-1 translocates to the nucleus and regulates the transcription of mitochondrial homeostasis related genes [149]. On the other hand, it has also been suggested that during UPR^mt^, general control non-repressed 2 (GCN-2) kinase gets activated by mitochondria reactive oxygen species (ROS) that act as a complementary pathway of HAF-1 and ATFS-1 [150]. Activated GCN-2 phosphorylate eukaryotic translation initiation factor 2α (eIF2α) and inhibit cytoplasm protein translation to decrease the folding load in the mitochondria [150].

In mammalian cells, the UPR^mt^ was first discovered with the deletion of mtDNA and overexpression of a deletion mutant of mitochondrial-targeted matrix protein ornithine carbamoyl transferase (ΔOTC) which resulted in transcriptional activation of nuclear-encoded mitochondrial chaperones HSP60, HSP10, and mtDnaJ, and the mitochondrial protease CLPP [144,151]. Furthermore, upregulation of other mitochondrial proteases, including YME1L1 and PMPCB, the import component TIMM17A, and the enzymes NDUFB2, endonuclease G, and thioredoxin 2 are also linked with UPR^mt^ induction [152]. Subsequently, various other mitochondrial dysfunctions and proteotoxic stress have been shown to induce UPR^mt^ that include loss of mitochondrial translation via deletion of DARS2 (mitochondrial aspartyl-tRNA synthetase), muscle-specific deletion of Crif, a large mitoribosomal subunit (39S) protein, and inhibition of matrix-localized HSP90/TRAP1 or LON protease [153,154,155]. Transcription factors required for the UPR^mt^ induction include CHOP (GADD153/DDIT3), C/EBPb, Jun, ATF5, and ATF4. CHOP was the first transcription factor reported for HSP60 upregulation in response to mitochondrial unfolded proteins [151]. However, CHOP can also be activated during ER stress (UPR^ER^) [156], questioning its specificity in UPR^mt^ regulation. The CHOP gene encodes a bZIP transcription factor that dimerizes with C/EBPb (CCAAT/enhancer-binding protein). Both CHOP and C/EBPb promoters showed the presence of an AP-1 (activator protein-1) site, which is specific for the induction of genes in UPR^mt^, and not in UPR^ER.^ [157]. Thereby, AP-1 presents context specificity for CHOP. AP-1 binds c-Jun, suggesting the contribution of the additional Jun transcription factor [157,158,159]. Evidence also suggests the requirement of JNK2 (cJun NH(2)-terminal kinase (JNK)), a kinase upstream of Jun, in CHOP induction but the signaling mechanism for JNK2 activation in UPR^mt^ is not known. Promoters with CHOP binding sites have two other conserved regions known as mitochondrial UPR elements (MURE1 and MURE2) that are known to get activated in mitochondrial stress conditions [152]. However, transcription factors interacting with MURE1 and MURE2 are not known.

After the discovery of CHOP, considerable efforts have been focused on the CHOP axis of the UPR^mt^. CHOP is reported to increase the expression of additional transcription factors that further amplify the stress response. More recently, CHOP-targeted transcription factor, ATF5 (the mammalian ortholog of ATFS-1) was identified [160]. ATF5 induced a similar transcriptional response and regulation in worms lacking ATFS-1, suggesting a similar function to *C. elegans* ATFS-1 [160]. ATF5 is particularly translated during the phosphorylation of the α subunit (at Ser^51^) of translation initiation factor 2 (eIF2α). The four eIF2α kinases including general control non-depressible 2 (GCN2 encoded by EIF2AK4), heme-regulated inhibitor (HRI encoded by EIF2AK1), PKR-like endoplasmic reticulum kinase (PERK encoded by EIF2AK3/PEK) and protein kinase R (PKR encoded by EIF2AK2)) mediate the phosphorylation of eIF2α (eIF2α∼P) [161]. These kinases are activated under different cellular stress responses [161]. The eIF2α∼P triggers suppression of global protein synthesis to prevent further organelle stress, while favoring translation of selected mRNAs. In mammalian cells, in addition to ATF5, eIF2α∼P increases the translation of ATF4 and CHOP itself that encode basic zipper (bZIP) transcription factors [162]. The eIF2α∼P/ATF4 pathway is known as the integrated stress response (ISR). In fact, ATF4 act upstream of CHOP and ATF5. Both ATF4 and CHOP, bind to ATF5 promoter and promote ATF5 transcription, which subsequently amplifies the transcription of mitochondrial stress response target genes [163].

Conversely, how mitochondria stress triggers the ISR and specificity of cytosolic eIF2α kinases in mitochondrial stress response was not clear. Recently, Guo et al. and Fessler et al. discovered the mechanism “known as OMA1–DELE1–HRI pathway” by which mitochondrial dysfunction communicates with ISR [164,165]. Both groups identified HRI, as necessary eIF2α kinase for ISR induction upon mitochondrial stress. Using a genetic screening strategy, the authors identified that mitochondrial proteostatic stress activated OMA1 protease, which cleaves the IMM protein DELE1. Cleaved DELE1 is translocated to the cytosol, where it interacts with the HRI to induce the eIF2α∼P, which subsequently initiates the ISR. These studies provide the missing link between mitochondrial stress and the ISR dependent activation of UPR^mt^ transcription factors. However, it is still not clear how cells cope with stress to promote cell death or survival; how this pathway works in disease conditions associated with mitochondrial dysfunction; whether the activation of this pathway in different disease conditions is protective or maladaptive; what other upstream and downstream signaling events occur; and whether and how a post-translational modification regulates ATF4, CHOP or ATF5. Further investigation of the UPR^mt^ pathway will offer a better understanding of this mechanism.

## 4. Conclusions and Future Directions

Mitochondrial protein homeostasis is vital for various physiological processes, and its dysregulation results in severe mitochondrial dysfunction. The mitochondrial proteome is not well characterized. Therefore, MPQC associated molecular events are not entirely known and still emerging. Recent discoveries of UPR^mt^, mPOS, UPRam, and ISR activation due to mitochondrial proteostatic stress suggest that MPQC systems not only affect the mitochondria, but also modulate the machinery of nuclear transcription and cytosolic protein synthesis. Thereby, the MPQC system represents an intricate network of retrograde-anterograde signaling between the mitochondria, cytosol, and nucleus. The protein half-life or turnover rate of the mitochondrial membrane-embedded proteins is longer compared to soluble proteins present in the matrix, IMS, and the cytosolic face of the OMM. Most of the soluble proteins are critical regulatory proteins involved in fine-tuning of the mitochondrial processes. Hence, the protein turnover of these soluble proteins represents an intricate physiological regulatory event. A tightly coupled molecular interplay between multiple players of the MPQCs ensures mitochondrial proteostasis and functionality to maintain overall cellular fitness. Most of the studies on the mitochondria quality control systems are performed using non-vertebrate model systems. These systems are largely conserved throughout evolution, but still, need to be studied in the mammalian system. The involvement of most of MPQC components in human pathologies and tissue/cell type-specific mitochondrial proteome and microproteome is entirely unknown and needs further investigation. Recently, multiple retrograde signaling events have been discovered, which are activated by the mitochondrial proteome in response to proteotoxic stress. Though, how precisely these signaling events parley with metabolism, cell death, and epigenetic regulatory events is not established. Molecular and physiological understanding of mitochondrial protein homeostasis and MPQCs is essential to understand the basic functioning of mitochondria and the cellular responses to dysfunctional organelles.

## Figures and Tables

**Figure 1 genes-11-00563-f001:**
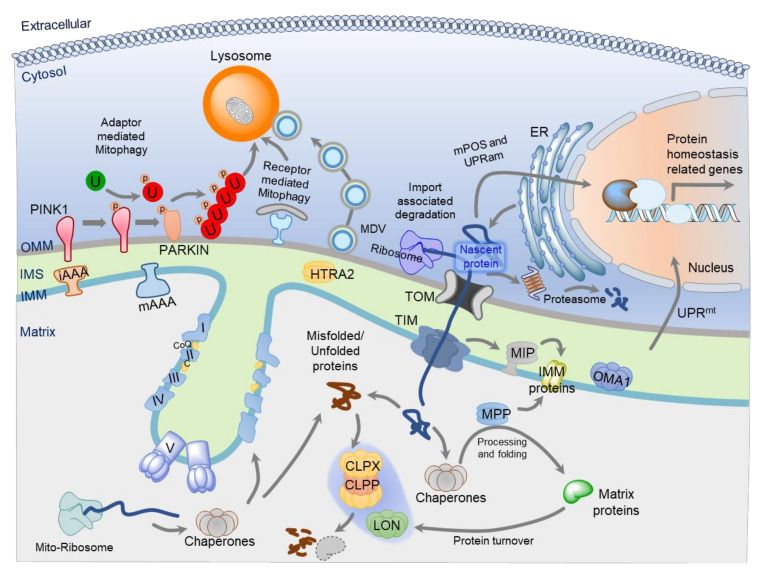
Overview of the Mitochondrial Protein Quality Control System (MPQC). Mitochondria maintain their protein-quality control at various steps and protect cells against proteotoxic stress. Most of the mitochondrial proteins are synthesized at ribosomes anchored to OMM or ER surface. Nascent mitochondrial proteins are imported into the mitochondria via the TOM/TIM translocase machinery. At the outer face of the OMM, mitochondria have proteostasis system (mPOS, UPRam, and UPS) to ensure that mitochondria maintain their full protein-import capacity. Mitochondrial protein import is assisted by the matrix localized Hsp70 chaperone. Upon import, chaperones together with proteases, ensure the proper trafficking and processing of nascent proteins. Both the nDNA or mtDNA encoded proteins fold into their native conformation, assisted by mitochondrial Hsp70 and Hsp60 chaperone systems. The targeting sequence is cleaved off by the matrix processing peptidase (MPP), giving rise to the mature protein. Some preproteins undergo additional processing by MIP after the removal of their presequence. Under stress or disease conditions, proteins become damaged or denatured. The damaged proteins can be removed by LON and CLPX/CLPP proteases in the matrix; mAAA and iAAA in IMM; and HTRA2 in IMS. Additionally, cells can transcriptionally upregulate the expression of mitochondrial chaperone and protease genes via a mechanism known as UPR^mt^ to relieve stress and re-establish homeostasis. In a failure of these MPQC mechanisms, cells have an alternative mechanism to remove damaged mitochondria. Dysfunctional mitochondria can be removed through the specific autophagy known as mitophagy, and part of mitochondria can be cleared through the mitochondrial-derived vesicles (MDVs). Mitophagy in mammals could be mediated either by a receptor or the PINK1/PARKIN-mediated pathway. Altogether, these mechanisms preserve mitochondrial quality. ER, endoplasmic reticulum; TOM, translocase of the outer membrane; TIM, translocase of the inner membrane; UPS, ubiquitin-proteasome degradation system; mPOS, mitochondrial precursor over-accumulation stress; UPRam, unfolded protein response activated by mistargeting of proteins; MDV, mitochondria-derived vesicles; PINK1, phosphatase and tensin homolog (PTEN)-induced kinase; PARKIN, E3 ubiquitin-protein ligase parkin; U, Ubiquitin; UPR^mt^, mitochondrial unfolded protein response; iAAA, inner membrane-embedded AAA protease; mAAA, matrix-embedded AAA protease; MIP, mitochondrial intermediate presequence protease; MPP, matrix processing peptidase; OMA1, overlapping activity with m -AAA protease; HTRA2, High-temperature requirement protein A2; CoQ, Coenzyme Q; LON, Lon protease; CLPX, ATP-dependent Clp protease proteolytic subunit X; CLPP, ATP-dependent Clp protease proteolytic subunit P; Mito-Ribosome, Mitochondrial Ribosome.

**Table 1 genes-11-00563-t001:** Compartmentalization of mitochondrial proteome, with the number of proteins and microproteins present in different mitochondrial compartments. OMM, outer mitochondrial membrane; IMM, inner mitochondrial membrane (IMM); IMS, intermembrane space.

Mitochondrial Compartment	Number of Proteins	Number of Microproteins
OMM	~140	1
IMS	~130	unknown
IMM	~800	3
Matrix	~500	unknown
Total	~1570	~25 (mostly uncharacterized)

**Table 2 genes-11-00563-t002:** Mitochondria associated degradation (MAD) or mitochondrial protein import associated degradation systems. VMS1, VCP associated Mitochondrial Stress-responsive 1; VCP, Velocin-containing protein; MDM30, Mitochondrial distribution and morphology protein 30; mitoRQC, Mitochondria associated ribosome quality control; MAGIC, mitochondria as guardian in cytosol; mPOS, mitochondrial precursor over-accumulation stress; UPRam, unfolded protein response activated by mistargeting of proteins; mitoCPR, mitochondrial Compromised Protein Import Response; mitoTAD, mitochondrial protein translocation-associated degradation; Cdc48, Cell division control protein 48; p97, 15S Mg(2+)-ATPase p97 subunit; Rqc2, Ribosome quality control complex subunit 2; NEMF, Nuclear export mediator factor; Ltn1, E3 ubiquitin-protein ligase listerin; HSP70, Heat shock 70 kDa protein; LONP, Lon protease homolog; Gis2, GIg Suppressor 2; Nog2, Nucleolar GTP-binding protein 2; Cis1, CItrinin Sensitive knockout 1; Msp1, Mitochondrial Sorting of Protein 1; ATAD1, ATPase family AAA domain-containing protein 1; Ubx2, UBX domain-containing protein 2.

MAD	Components
VMS1/VCP/MDM30	VMS1, Cdc48/VCP/p97, MDM30 (SCF ubiquitin ligases)
mitoRQC	VMS1, Rqc2/NEMF, Ltn1/Listerin
MAGIC	Cytosolic HSP70s, LONP
mPOS	Ribosome associated proteins Gis2, Nog2
UPRam	Downregulated ribosomal and upregulated proteasome protein expression
mitoCPR	Cis1, Msp1/ATAD1
mitoTAD	Ubx2, Cdc48/VCP/p97

**Table 3 genes-11-00563-t003:** Mammalian resident mitoproteases involved in MPQC. Oct1, Octapeptidyl aminopeptidase 1; Mas2, Mitochondrial Assembly protein 2; Mas1, Mitochondrial Assembly protein 1; Mop112, Metalloprotease of 112 kDa; Qri7, tRNA N6-adenosine threonyl carbamoyl transferase; Map1, Methionine aminopeptidase 1; Icp55, Intermediate cleaving peptidase 55; Pim1, Proteolysis In Mitochondria; Yme1, Yeast Mitochondrial Escape; Yta10, Yeast Tat-binding Analog 10; Yta12, Yeast Tat-binding Analog 12; Oma1, Overlapping activity with M-AAA protease; Cor1, Core protein of QH2 cytochrome c reductase; Ocr2, QH2:cytochrome-C oxido Reductase; Imp1, Inner Membrane Protease 1; Imp2, Inner Membrane Protease 2; Pcp1, Processing of Cytochrome c Peroxidase 1; Prd1, Proteinase yscD1; Ubp16, Ubiquitin-specific Protease 16.

Protease	Localization	Catalytic Class	Yeast Orthologue
Mitochondrial intermediate peptidase (MIP)	Matrix	Metallo	Oct1
Mitochondrial-processing peptidase **subunit α** (PMPCA)	Matrix	Metallo	Mas2 (MPP)
Mitochondrial-processing peptidase **subunit β** (PMPCB)	Matrix	Metallo	Mas1 (MPP)
Presequence protease 1 (PITRM1)	Matrix	Metallo	Mop112
O-sialoglycoprotein endopeptidase-like protein 1 (OSGEPL1)	Matrix	Metallo	Qri7
Methionine aminopeptidase 1D (METAP1D)	Matrix	Metallo	Map1
Xaa-Pro aminopeptidase 3 (XPNPEP3)	Matrix	Metallo	Icp55
Parkinson disease protein 7 (PARK7)	Matrix	Cysteine	Unknown
ATP-dependent Clp protease proteolytic subunit (CLPP)	Matrix	Serine	Unknown
Lon protease-like protein (LONP)	Matrix	Serine	Pim1 (Lon)
YME1-like protein 1 (YME1L1)	IMM	Metallo	Yme1 (i-AAA)
Spastic paraplegia 7 protein (SPG7)	IMM	Metallo	Yta10 (m-AAA)
AFG3-like protein 2 (AFG3L2)	IMM	Metallo	Yta12 (m-AAA)
Overlapping with the m-AAA protease 1 homolog (OMA1)	IMM	Metallo	Oma1
Ubiquinol-cytochrome-c reductase complex core protein 1 (UQCRC1)	IMM	Metallo	Cor1
Ubiquinol-cytochrome-c reductase complex core protein 2 (UQCRC2)	IMM	Metallo	Qcr2
Mitochondrial inner membrane protease subunit 1 (IMMP1L)	IMM	Serine	Imp1 (IMP)
Mitochondrial inner membrane protease subunit 2 (IMMP2L)	IMM	Serine	Imp2 (IMP)
Presenilins-associated rhomboid-like protein (PARL)	IMM	Serine	Pcp1 (Rhomboid)
Mitochondrial inner membrane protease ATP23 homolog (ATP23)	IMS	Metallo	Atp23
Microsomal endopeptidase (MEP) or Neurolysin	IMS	Metallo	Prd1
High-temperature requirement protein A2 (HTRA2) or OMI	IMS	Serine	Unknown
Serine β-lactamase-like protein LACTB (LACTB)	IMS	Serine	Unknown
Ubiquitin-specific-processing protease 30 (USP30)	OMM	Cysteine	Ubp16

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
