# Peer review of "Mitochondrial Protein Quality Control Mechanisms"

_genes, 2020, doi:10.3390/genes11050563_

Round 1
Reviewer 1 Report
Authors provided a brief overview of mitochondrial quality control. The paper is overall well organized, but a few changes should be made.
1.There are several grammatical mistakes throughout the manuscript. Authors should have a native English speaker review the manuscript before resubmission. A few examples are listed below:
A. Line 65: Authors should not use "we think that" in the review.
B. Line 88, 95, and many other places in the review, authors should change "evidences suggest" to "evidence suggests"
C. Line 149: remove "the" from "the mitophagy ".
2. Authors discuss parkin in several places before Parkin/mitophagy is described. Some reorganization of the review is suggested.
3. Authors should move figure 1 to the end as a summary figure.
4. Line 448, authors say there are 5 adaptors proteins, but only 4 are mentioned.
Author Response
Reviewer 1:
Authors provided a brief overview of mitochondrial quality control. The paper is overall well organized, but a few changes should be made.
Response: We greatly appreciate the positive response. The manuscript is revised according to the reviewer suggestions.
1.There are several grammatical mistakes throughout the manuscript. Authors should have a native English speaker review the manuscript before resubmission. A few examples are listed below:
Response: We apologize for the grammatical mistakes. Revised manuscript is proofread by one of our colleagues specifically for the language, grammar, and typos.
- Line 65: Authors should not use "we think that" in the review.
Response: Thanks for highlighting this. We revised the sentence as "Experimental evidence suggests that dysfunctional mitochondrial protein quality control is an antecedent cellular event involved in several human diseases."
- Line 88, 95, and many other places in the review, authors should change "evidences suggest" to "evidence suggests"
Response: Thanks for the suggestion. These typos are corrected throughout the manuscript.
- Line 149: remove "the" from "the mitophagy ".
Response: We corrected it.
- Authors discuss Parkin in several places before Parkin/mitophagy is described. Some reorganization of the review is suggested.
Response: Thanks for pointing this out. Parkin is a ubiquitin E3 ligases and involved in ubiquitination and turnover of multiple outer mitochondrial membrane proteins, besides its conventional role in mitophagy. Therefore, it described in several places. We discussed Parkin in the context of mitophagy where its role is well studied, but we also highlighted its involvement in other protein quality control systems, which is unavoidable.
- Authors should move figure 1 to the end as a summary figure.
Response: We appreciate the reviewers' suggestion. However, Figure 1 represents a broad overview of the mitochondrial protein quality control mechanisms instead of a summary. Therefore, Figure 1 makes more sense in its current position in the manuscript.
- Line 448, authors say there are 5 adaptors proteins, but only 4 are mentioned.
Response: We appreciate the reviewer for pointing this out and apologize for missing the fifth adaptor protein. Now we included the "nuclear domain 10 protein 52 (NDP52)".
Reviewer 2 Report
This manuscript by Pooja Jadiya and Dhanendra Tomar reviews the Mechanisms of Mitochondrial Protein Quality Control seems to have relevant insights to treat multiple human diseases. The content is interesting, the paper is well written, and the science behind it has been previously validated through peer-review. I only have a few recommendations that should be clarified before publication:
- Line 65… “inadequate” … please change it to “insufficient” or another synonym
- Line 90… “inadequate” … please change it to “incomplete” or another synonym
- Lines 106-107…“.. involved in electron transport named ETC complexes I to V (C-I, C-II, C-III, C-IV, C-V) ”… “please change it to involved in electron transport named ETC complexes I to IV (C-I, C-II, C-III, C-IV) and ATP synthase”
- Line 221 … The VCP associated – Is it Valosin-containing protein? Please explain
- I suggest that for a better understanding of the material include schemes reflecting the described processes. For example, paragraph 3.4, line 454, the authors write that the process is well studied.
- Figures and tables should be where the first mentioned in the text, move please table 1
- The manuscript requires the same design (for example, line spacing).
- I would like the conclusions to be clearer. it is likely that there are a lot of unclear and unconfirmed points in the question under study, however, the authors could make several assumptions based on the material worked out
Author Response
Reviewer 2:
This manuscript by Pooja Jadiya and Dhanendra Tomar reviews the Mechanisms of Mitochondrial Protein Quality Control seems to have relevant insights to treat multiple human diseases. The content is interesting, the paper is well written, and the science behind it has been previously validated through peer-review. I only have a few recommendations that should be clarified before publication:
Response: We are thankful to the reviewer for appreciating our work.
Line 65… "inadequate"… please change it to "insufficient" or another synonym
Line 90… "inadequate"… please change it to "incomplete" or another synonym
Response: We appreciate the reviewer comments. At the line 65, we rephrased the sentence for more clarity and included the reviewer suggestion. Line 65 is now read as “However, the understanding of these pathways in human physiology and their association with pathologies remains insufficient.” Line 90 changed according to reviewer suggestion.
Lines 106-107… ".. involved in electron transport named ETC complexes I to V (C-I, C-II, C-III, C-IV, C-V)"… "please change it to involved in electron transport named ETC complexes I to IV (C-I, C-II, C-III, C-IV) and ATP synthase"
Response: Thank for pointing this out. We revised it.
Line 221 … The VCP associated – Is it Valosin-containing protein? Please explain
Response: Yes, it is Velocin-containing protein, we included the full-form to avoid any confusion.
I suggest that for a better understanding of the material include schemes reflecting the described processes. For example, paragraph 3.4, line 454, the authors write that the process is well studied.
Response: We appreciate the reviewer comment. There are multiple outstanding review articles available which covers the involvement of the PINK1/Parkin in mitophagy and we do not want to repeat the content. These reviews included outstanding schemes to define the PINK1/parkin role in mitophagy (For e.g.: Mechanisms of mitophagy: putting the powerhouse into the doghouse by Riley and Tait, JBC 2016; PINK1 and Parkin: emerging themes in mitochondrial homeostasis by McWilliams and Muqit, Current Opinion in Cell Biology, 2017; Mitophagy and Quality Control Mechanisms in Mitochondrial Maintenance by Pickles, Vigié, and Youle, Current Biology, 2018). Therefore, we have not provided the scheme for this part.
Figures and tables should be where the first mentioned in the text, move please table 1
Response: Thanks for pointing this out, we moved the table appropriately.
The manuscript requires the same design (for example, line spacing).
Response: We corrected it.
I would like the conclusions to be clearer. it is likely that there are a lot of unclear and unconfirmed points in the question under study, however, the authors could make several assumptions based on the material worked out
Response: We appreciate the reviewer suggestion. We have revised the conclusion for more clarity.